# Decoding Human Preferences in Alignment: An Improved Approach to Inverse Constitutional AI

## Abstract

Traditional methods for aligning Large Language Models (LLMs), such as Reinforcement Learning from Human Feedback (RLHF) and Direct Preference Optimization (DPO), rely on implicit principles, limiting interpretability. Constitutional AI (CAI) offers an explicit, rule-based framework for guiding LLM alignment. Building on this, we refine the Inverse Constitutional AI (ICAI) algorithm, which extracts constitutions from preference datasets. By improving principle generation, clustering, and embedding processes, our approach enhances the accuracy and generalizability of extracted principles across synthetic and real-world datasets. Our results highlight the potential of these principles to foster more transparent and adaptable alignment methods, offering a promising direction for future advancements beyond traditional fine-tuning.

## 1 Introduction

Multiple options exist to align pre-trained Large Language Models (LLMs) to better adhere to human preferences. Popular methods include Reinforcement Learning from Human Feedback (RLHF), which trains a reward model to act as a proxy for human feedback to rate model outputs, and Direct Preference Optimization (DPO), which eliminates an explicit reward model to represent human preferences, and instead, implicitly defines this in their loss function for fine-tuning. Both approaches heavily rely on pairwise human-annotated preference data that rank model outputs.

As an alternative to traditional techniques, Anthropic introduced Constitutional AI (CAI) (Bai et al., 2022), which offers a rule-based alignment based on a core set of principles/values called the constitution. This set contains key ethical, moral, and safety standards that guide the outputs and promote desired behaviors through repeated critiquing of model outputs. Having an explicitly defined set of core values aids in the interpretability of the changes induced through the alignment procedure, as typical approaches like DPO or RLHF rely on an implicitly defined set of principles embedded in the pairwise preference data. The latter approaches introduce difficulty due to the potential inclusion of majority group biases Sorensen et al. (2024); Chakraborty et al. (2024). Inclusion of explicit goals or rules is also a well motivated philosophical approach, which several works believe to be the best fit to the core idea of alignment Ji et al. (2025); Gabriel (2020); Tennant et al. (2025).

---

*These authors contributed equally to this work.

Building on the idea of CAI, Findeis et al. (2024) proposed an Inverse Constitutional AI (ICAI) algorithm. The algorithm extracts a constitution from a pairwise preference dataset through a multistep process involving prompting, clustering, and LLM-as-a-judge feedback. Through this process, we gain insights into the core values represented in a dataset, ranging from style and ethical values to content preferences. We hypothesize that the produced constitutional principles could then be used in a prompting-based manner to steer outputs to follow the principles, i.e., produce pseudo-aligned outputs.

Our work aims to improve the shortcomings that we identified within the ICAI algorithm to produce constitutional principles that represent dataset preferences more accurately. For this purpose, we address different weaknesses regarding generalizability and sampling. We also experiment with utilizing various embeddings to perform grouping of related preference pairs prior to the initial principle generation. We evaluate our changes in three settings, ranging from synthetic to semi-synthetic and realistic data, and report improvements over the baseline ICAI algorithm in all three.

Overall, our work aims to tackle the following research questions:

1. How can we improve the existing constitutional extraction method on pair-wise preference human datasets?

2. What are the key use cases and implications that such representative constitutions provide us with?

3. Can the generated constitutions constitute a competitive alternative to traditional fine-tuning methods?

## 2 Methodology

Findeis et al. (2024) utilize a combination of prompting, clustering, and voting to extract a representative set of rules. The ICAI algorithm takes a pairwise preference dataset as input and performs the following five steps to derive a constitution:

1. **Initial candidate generation**: Using a pair of chosen and rejected replies, the algorithm prompts an LLM to generate a set of candidate principles that reflect the preference rating.

2. **Clustering**: All principles are embedded and clustered using KMeans.

3. **Subsampling**: A random principle is chosen from each cluster to represent a candidate for the final constitution.

4. **Testing**: Using an LLM, each candidate principle is evaluated against every pair in the preference dataset. Ratings are collected on whether a candidate is in favor/against/not applicable to each preference pair.

5. **Filtering**: Finally, based on a set of rules/thresholds, the list of candidate principles is reduced to the final constitution.

We first analyze the implementation of the ICAI algorithm and identify possible improvements in multiple steps. From the architecture, it is clear that the pipeline heavily relies on the ability to

generate representative and generalizable principles from a single preference pair, which is rooted in step one. This essentially constitutes a bottleneck since rules are not changed in later stages; they are only filtered. This is problematic since our test runs of the pipeline revealed that the principles were often heavily tailored to the specific sample they were derived from. For example, "Select the response that engages with the user's interest in Sahawiq." was one of the identified principles (Sahawiq being a hot sauce). Because of such candidates, we believe limiting effects in later processing steps reduce overall effectiveness.

Further, in step three, a principle is randomly selected from the resulting clusters. As described above, some principles are highly specific and should not be considered; however, principles of such sub-par quality may still be selected in this step and misrepresent the cluster contents. Overall, we believe that the ICAI architecture suffers from the propagation of weak, non-representative principles into late processing steps, which replace promising alternatives. Additionally, the model heavily relies on its initial principles, which require drafting a large number to ensure stable results. This is a bottleneck concerning result quality and incurs unnecessary performance overhead.

Our proposed improvements address these issues. First, we modify the principle generation prompt to nudge the LLM to generate a set of more generalizable principles. Secondly, during principle sub-sampling, our approach selects the principle from the cluster closest to its centroid (as measured by the cosine similarity of embeddings). These two joint improvements result in our first improved version.

We also explore further enhancements aimed at improving the quality of the initial principal candidates. This step is based on the assumption that the rule we seek to extract is reflected in the difference between the two responses. This difference can potentially be represented as a tensor in latent space, provided we have an appropriate representation of the responses. We further believe that different rules may target different aspects of alignment, including style ("Speak in a kind and concise manner"), content ("Do not give information on illegal activities"), and sentiment ("Do not agree on problematic claims and try to resolve them"). In line with these assumptions, we employ different embedding models to produce the needed representation of pairs and consequently their difference. In particular, we employ all-mpnet-base-v2 (Henderson et al., 2019) for content, Wegmann et al. (2022) for content-independent style, and Sanh et al. (2020) for sentiment classification.

Using these models, we arrive at three embedding difference maps and use KMeans to identify clusters, which we hope will feature a joint principle. To consolidate the best candidates from every map, we compute a combined score of inter- and intra-cluster distance and only extract the top k clusters over all three. We also ensure no overlaps between nodes in different chosen clusters. Since some clusters tend to incorporate a large number of nodes, we refine the extraction further by focusing on node triplets within a cluster. We reuse the distance metrics from the cluster selection to choose representative triplets. In this step, we aim to ensure a rule majority in cluster representations, which we validated on a synthetic dataset, where the top 5 triplets consistently achieved a purity of at least 66.7%. Since the underlying data is synthetic, this value can be computed by inspecting the ground truth principle used to generate the preference pair. These triplets are then used in a joint prompt to generate initial candidates and complete the second improved version of the pipeline. We illustrate this in Figure 1.

We evaluate the baseline and both improved versions in three settings:

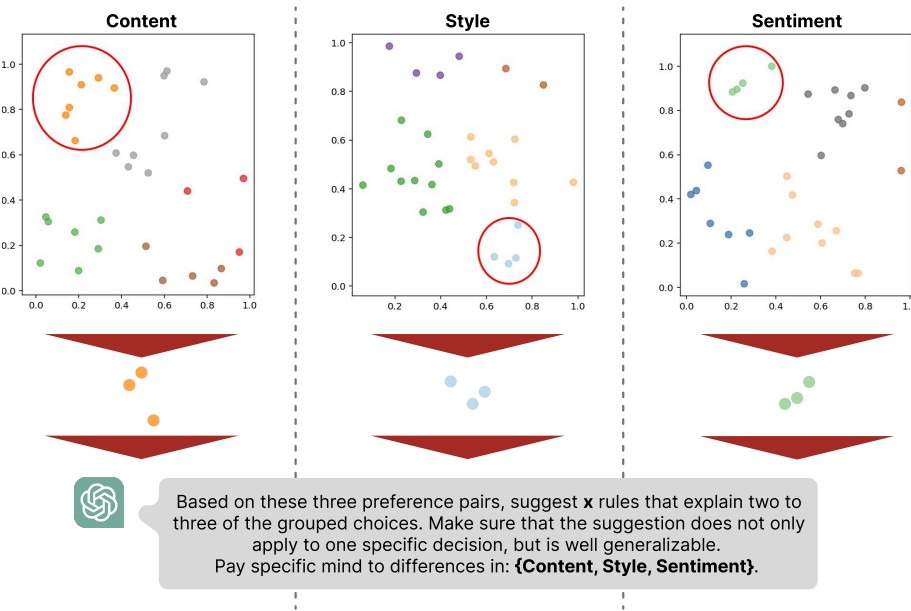

Figure 1: We apply KMeans on the difference between the embeddings of the preference pairs. After estimating cluster potential, we extract representative node triplets, which are inserted in one joint principle generation prompt. Prompts are executed separately for each dimension.

1. A synthetic setting where we explicitly control for the preferences elicited in the pairwise preference data.

2. A semi-synthetic setting derived from pairs that elicit significant differences in scored preference.

3. A realistic/original setting where we sample principles from a pairwise preference dataset without intervention.

The synthetic dataset is generated by choosing five constitutional principles from the original CAI paper (Bai et al., 2022). Next, we sample 150 pairs of chosen and rejected outputs from Anthropic's HH dataset's harmlessness subset. We keep the rejected output and prompt GPT-4o to re-write the output based on one of the constitutional principles, obtaining a synthetic chosen output. The chosen principles include an explicit rewriting prompt, which we utilize for this purpose. Each principle is represented equally among the train set (100) and test set (50). The process is visually described in Figure 2.

To obtain a semi-synthetic dataset, we choose to filter a subsample of data points from the Ultra-Feedback (Cui et al., 2024) dataset, which conveniently includes ratings (ranging between 1 and 5) for the rejected and chosen outputs. We filter to only include samples where the rating difference between chosen and rejected is at least two and then, using weighted sampling based on the distances, sample 1000 data points for the train set and 500 for the test set. We display the corresponding process in Figure 3. We apply this filtering since outputs with a similar rating according

to the annotator's preference do not carry much useful information for extracting an underlying rule according to the original ICAI algorithm and our improved versions. On the contrary, very slight differences may lead to the extraction of unwanted or incorrect rules. However, we believe this may change when the scores are an active part of the extraction algorithm, which we investigate in section 3.

Lastly, our original/realistic setting consists of a subset of the HH-Harmlessness dataset of 1000 pairs for the train set and 500 for the test set. We note that the HH dataset includes very subtle differences between the replies, which makes it a challenging target, so much so that even humans struggle to infer preferences when the labels are removed.

For our main evaluation, we use an LLM-as-a-judge approach and iterate over all samples in the test set, prompt the LLM with the chosen and rejected sample, include all constitutional values, and ask the model to choose the output that aligns best with the provided principles. We believe that regenerating the preferences is a natural measure of the quality of our approximation of the underlying latent preference set. We adjust for ordering bias by gathering scores in both orders for each pair and then averaging the two scores.

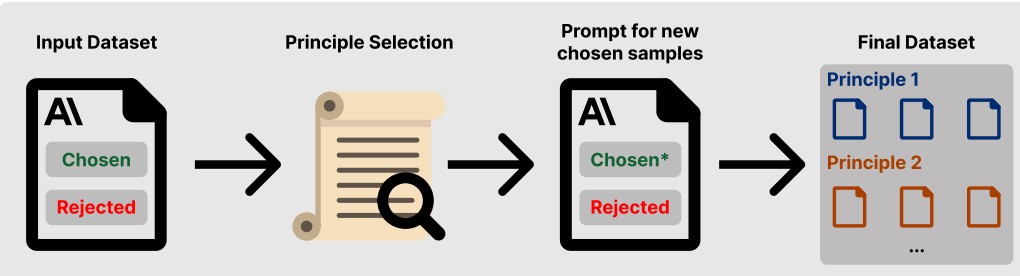

Figure 2: Dataset generation process for the synthetic dataset. Colors in the final dataset represent the samples that were generated using the same principle.

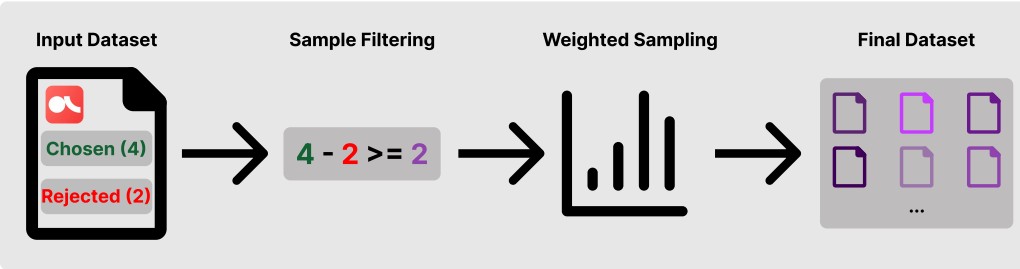

Figure 3: Dataset generation process for the semi-synthetic dataset. Different shades of purple in the final dataset indicate different deltas between chosen and rejected ratings.

## 3 Experimental Results

### 3.1 Regenerating Preferences

Our experimental results on the regeneration of preferences in the three different settings are outlined in Table 1. We include an orthogonal approach (unrelated constitutional values, such as prefer

| Dataset | Baseline | Orthogonal | Improved 1 | Improved 2 |
|---------|----------|------------|------------|------------|
| Synthetic | 92.00% | 62.50% | **94.00**% | 93.00% |
| Semi-Synthetic | 71.20% | 46.95% | 73.80% | **76.20**% |
| Original | 60.65% | 56.60% | 60.55% | **60.75**% |

Table 1: We report test-set accuracy as measured by how many sample preferences can be regenerated using the extracted constitution. Our results show slight improvements in the synthetic and realistic setting and significant improvements in the semi-synthetic setting.

cats over dogs) to show reference win rates and control for inherent model bias. For this purpose, we adapt the orthogonal constitution used in Findeis et al. (2024). The orthogonal constitution is held constant for all three datasets. Specifically, for the latter two datasets, we see accuracies of around 50%, which we would expect, as this represents a random choice. The accuracy for the synthetic dataset is slightly higher; however, accuracies for all other approaches outperform it significantly. We hypothesize that the harmful character of the rejected prompts makes it difficult for the model to choose them despite being instructed to abandon judgment beyond the constitutional rules.

### 3.2 Constitution Similarity

To evaluate how our generated constitutions compare to the ground truth constitution, we conduct an experiment employing LLM-as-a-judge to estimate similarity. For this purpose we first estimate similarity values between candidates and ground truth and create an optimal matching. After that, we aggregate the similarity scores. The results are shown in Table 2.

We also measure the ground truth constitution against itself. As the scoring scale ranges from 1 to 10, an average score of 5.8 is significantly lower than expected, raising concerns about the LLM's ability to correctly match and rate between constitutional values. This may also partially be due to a different output format ("choose the response that", and "Explain the prompt so that"), which the model is instructed to disregard for scoring. To correctly interpret the other results, we

| Approach | Similarity Score |
|----------|------------------|
| Ground Truth | 5.8 |
| Baseline | 5.0 |
| Orthogonal | 2.2 |
| Improved 1 | 5.0 |
| Improved 2 | **5.4** |

Table 2: We report mean similarity scores (1-10) between the ground truth constitution and the constitution generated using the synthetic dataset (the only setting in which a known ground truth constitution exists). Similarity scores are averaged across all constitutional values (n=5).

compare values to the ground truth score of 5.8 rather than the max score of 10. Our results show a significant improvement in our Improved 2 approach over the baseline. Again, we include the orthogonal constitution to show that other scores differ significantly.

### 3.3 Embedding Model Ablation

| Model Combination | Synthetic | Semi-Synthetic (UF) | Realistic (HH) |
|---|---|---|---|
| Content | 0.910 | 0.690 | 0.600 |
| Content + Style | 0.910 | **0.731** | 0.595 |
| Content + Sentiment | 0.910 | 0.723 | 0.598 |
| Style + Sentiment | 0.910 | 0.712 | 0.577 |
| **All (Full System)** | **0.920** | 0.723 | **0.608** |
| *Improvement (All vs Best)* | *+0.010* | *−0.008* | *+0.008* |

Table 3: Average scores across three evaluation datasets: Synthetic, Semi-Synthetic (UltraFeedback), and Realistic (Helpfulness-Harmlessness). Content refers to `all-mpnet-base-v2`, Style to `AnnaWegmann/Style-Embedding`, and Sentiment to `distilbert-base-uncased-finetuned-sst-2-english`. "All" includes all three models. The improvement row compares the full system to the best-performing preceding combination in each setting.

To better understand the contribution of each embedding model in *Improvement Two*, we conduct an ablation study with all combinations of the three embeddings. The results are shown in Table 3. For this analysis, we employ GPT-4.1-nano for all generation steps to produce constitutions conditioned on each embedding combination. Subsequently, we use GPT-4o to annotate a randomly subsampled set of preference pairs from the test set, evaluating alignment with the generated constitutions.

Due to the random subsampling and the use of a smaller model (GPT-4.1-nano), absolute scores are generally lower than in the full evaluation reported earlier in Table 1. Nonetheless, relative differences remain informative. We find that using all three embeddings consistently improves over the base Content-only system across all settings. In the Synthetic and Realistic settings, the full system achieves the highest overall scores. However, in the Semi-Synthetic setting, the combination of Content and Style outperforms the full system, suggesting that the addition of Sentiment embeddings may introduce noise or redundancy in that context.

These findings align with prior observations: the Content embedding often provides the most substantial lift, which is frequently enhanced by the addition of Style. Sentiment, while generally contributing modest gains, can sometimes slightly reduce performance depending on the dataset. Overall, although differences between combinations are relatively small, leveraging all three embeddings proves consistently beneficial compared to using Content alone.

### 3.4 AlpacaEval

In order to measure how useful extracted constitutions are for alignment purposes in a naive manner, we conduct an experiment using AlpacaEval (Li et al., 2023; Dubois et al., 2024). We sample

a test set of 100 instructions from the Ultra Feedback dataset (different than the one used in Preference Regeneration) and generate base responses using LLama3.1-8B-Instruct. We note that this model already possesses some basic alignment, meaning this experiment may underestimate the effect of including the constitution in the prompt. Further, constitutions do not single out a specific alignment aspect such as instruction-following, safety, or tone of voice but instead address multiple of these, meaning AlpacaEval is a suboptimal evaluation measure, as it is mainly focused on instruction-following. Nevertheless, they constitute a suitable preliminary evaluation technique to investigate the effects of prompt manipulation. We prepend the constitution and instruct the model to consider that its reply will be evaluated using these rules. We consider the Base constitution extracted from the synthetic dataset (5 rules) and the constitution extracted from Ultra Feedback (10 rules). The resulting win rates against the Llama version without the prompt manipulation are displayed in Table 4. We use the annotator configuration `alpaca_eval_clf_cot_gpt4_turbo`.

| Model | Win Rate (%) | Standard Error | Token Difference Median | LC-Win Rate (%) |
|---|---|---|---|---|
| Base_Con | 46.74 | 5.23 | -46.50 | 52.74 |
| UF_Con | 46.74 | 5.23 | -65.50 | 48.46 |

Table 4: AlpacaEval Results of LLama3.1-8B-Instruct with Base constitution and UF constitution evaluated against the version without. Token Difference median refers to the difference between the base model and the model using the constitution (generally produces shorter replies). LC = Length Controlled.

Eight instructions are excluded since no clear preferences were exhibited. Generally, results are mixed, with lower performance on the original evaluation metric and better or equal performance on the Length-Controlled version. Interestingly, both constitution sets seem to decrease output length. Most likely, this is due to rules that include clarity and/or conciseness, which are present in both constitutions. Despite being extracted directly from the UF dataset, we find that the constitution does not improve over the base constitution from the synthetic dataset. On the contrary, the synthetic constitution performs notably better on the length-controlled evaluation and outperforms the base model. A more comprehensive evaluation is required to fully explore the potential of in-context tuning with constitutions, which we plan to address in future work as resources permit. Based on our limited evaluation, it is unclear whether constitution-based prompting provides a viable alternative to traditional fine-tuning methods.

## 3.5 Scored Preference Datasets

Some alignment datasets, such as UltraFeedback, also include numerical ratings of both outputs of each preference pair. We hypothesize that these ratings could be employed in our pipeline to further improve the extraction of principles, as they contain valuable information about the extent to which a rule expresses a preference for one reply over the other. We slightly modify our first improved algorithm to test our hypothesis and include the UltraFeedback ratings in the initial principle generation prompt. We consciously do not adjust the rest of the algorithm to gain a sense of the performance improvement the model gains by employing the scores in the prompting step. Interestingly, this simple modification led to an accuracy of 76.80%, or an improvement of 3%, even slightly higher than our Improved version 2. We thus conclude that preference ratings bear significant potential for proper constitution extraction. Even without functional adaption of the

algorithm, the model is able to extract valuable information. Further optimization in this direction will likely enable us to extract an even more representative constitution in future work.

## 4 Related Work

Although the key relevant bodies of work are outlined in more detail in section 1, we provide further details on adjacent works below.

**Alignment Approaches**: Inclusion of explicit rules has been adapted in several alignment techniques. Klingefjord et al. (2024) include the generation of relevant alignment targets directly by a process they refer to as Moral Graph Elicitation, which includes an explicit interviewing and reconciliation stage with annotators. Glaese et al. (2022) set out to steer the underlying applied preference exhibited by annotators through providing pre-defined rules, which was used in combination with a traditional RLHF approach. Partially inspired by this, Anthropic proposed CAI (Bai et al., 2022), an alignment approach that instills constitutional principles into model outputs. Findeis et al. (2024) presented the Inverse CAI (ICAI) algorithm to extract a set of principles from a pairwise preference alignment dataset. Kostolansky (2024) proposes a similar approach to ICAI that also leverages clustering of embeddings and prompting-based steps.

Beyond the approaches centered around constitutional values for alignment / extracting these values, alternative alignment approaches exist that are tangent to constitutional ones. One is Dromedary (Sun et al., 2023), a self-alignment approach that relies on 16 guiding principles throughout their pipeline. Their solution aims to minimize the number of required human annotations. Gao et al. (2024) propose a framework, PRELUDE, which uses user edits to infer latent preferences, enabling alignment without costly fine-tuning.

Although Dromedary and PRELUDE may appear as competitive alternatives to ICAI, they may in fact be used in a synergetic manner. In Dromedary, the Self-Instruct stage requires human-written instructions to elicit the wanted replies, upon which the engraving (SFT) stage may follow. Although they are initially short and not very descriptive, it is possible to use the extracted principles for this purpose. One strategy could be to have a language model expand on them, while maintaining the instruction-phrasing. By using this, it is possible to apply Dromedary in combination with a human-annotated preference dataset, for example to benchmark against more native solutions. We note, however, that Dromedary cites the ability to circumvent such expensive annotation datasets as one of their key strengths, so this is likely more of an edge case. PRELUDE offers an interesting new target for the application of the improved ICAI algorithm, by sourcing preference pairs directly through user edits. Gao et al. (2024) also look into utilizing Language Models to explain the difference between user edits if they are considered costly, however, the strength of the ICAI approach lies in the aggregation of multiple such preferences over a dataset. Thus, it is difficult to directly incorporate in the context of PRELUDE. In addition, there are certain additional mechanisms, like retrieving relevant past edits, which are beyond the scope of our work.

**Value-based Methods**: Value-based approaches are not limited to language model alignment. Hosking et al. (2024) argue that human preferences are inherently multi-dimensional rather than one-dimensional. Their experiments reveal that multiple factors contribute to these preferences, with the refusal to answer unreasonable requests emerging as the most crucial. Similarly, Ji et al. (2023) introduce the BeaverTails dataset, a 330k-entry QA dataset annotated for helpfulness and

harmlessness. The harmlessness labels are further divided into 14 distinct categories of harmful values, providing a more nuanced framework for evaluating responses.

## 5 Conclusion

In our work, we explored a number of ways to improve on the recently proposed Inverse Constitutional Artificial Intelligence algorithm. We further investigated the implications of the extracted constitutions and their use cases. For this purpose, we employed datasets of varying syntheticity and multiple evaluation metrics. We find that our changes improve upon the base algorithm in all settings, albeit to varying extents. Commonly employed preference datasets with very subtle differences or differences that are difficult to interpret remain a significant challenge. The resulting constitutions are human-readable and naturally encode relevant preference information about the dataset, making them a strong interpretability artifact. However, their direct in-context employment does not yield clear performance increases in instruction-following evaluations when using AlpacaEval. We note that this is only a preliminary insight based on limited evaluation, and more sophisticated approaches may be able to utilize the constitution more effectively. Due to the filtering process during the constitution extraction, related techniques might have the potential to avoid learning unwanted preferences, such as producing longer outputs. The base constitution hints at such capabilities by improving the Length-Controlled Win Rate. We also discover potential in utilizing preference scores, which, when included in the prompt, led to an improvement in preference regeneration capabilities.

### 5.1 Discussion

Despite the improvements in preference regeneration, our approaches come with some downsides. The embedding and clustering process of improvement 2 poses a major drawback of the solution. The algorithm should be highly scalable as we want to obtain generalizable sets of constitutional principles from full relevant datasets. However, significant latency is introduced when embedding all preference pairs three times and clustering the results. We go further into runtime statistics regarding this in A. In addition, the base algorithm is already not very scalable as it sequentially prompts an LLM for every preference pair twice throughout the generation process. This incurs significant resource costs. A possible solution is to to parallelize these operations over the dataset, which Findeis et al. (2024) use as well, however, this may lead to contamination in the model replies. We believe that improvement 2 may actually constitute a step in the right direction if one falls back on a single embedding model and increases the number of chosen top clusters. In this case, parallel prompting is desirable since candidate rules should apply to all three given preference pairs.

Beyond efficiency, there are also limitations to the depth or ambiguity of extracted rules. Sometimes, annotators will prefer exhaustive, lengthy replies for certain instructions, such as coding tasks, but at other times they will prefer more concise replies. Extracting rules that depend on the given instruction is currently not possible and may pose further challenges due to inherent ambiguity.

### 5.2 Future Work

There are several directions in which this work could be expanded. We have already identified several weaknesses of current approaches, such as scalability, weak performance on the AlpacaEval evaluation suite, and unused preference scores. Potential topics could encompass sophisticated

approaches to use constitutions for alignment or the employment of preference scores directly in the extraction algorithm beyond only adapting the prompt. Apart from these topics, we consider whether this technique could also be applied to datasets that do not contain binary preferences but only positive or negative examples. Namely, we hypothesize that such demonstrations also contain common rules that can be inferred from multiple combined samples. Alternatively, the extraction might also be possible if the outputs contain a quality score. Another interesting direction would be the application of models serving as annotators. Given a binary dataset, eliciting preferences from a given model could yield a preference dataset which, after extracting a constitution, could explain the learned preferences of the model. This is especially interesting with regard to bias detection. In summary, we see multiple promising avenues for future work that either directly or indirectly address the limitations of our improvements.

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

## A  Resource usage for Improvement 2.

We profiled the full embedding–clustering pipeline node on our HPC system, using a dual-socket Intel Xeon machine with 32 physical cores and 384 GB RAM; one NVIDIA A100-40GB GPU was allocated for the run. The training split of 19,874 dialogue pairs (39,748 utterances) was processed end-to-end in 0.49 GPU-hours ($\approx$ 29 min wall-clock). Peak GPU memory reached 3.7 GB (queried via `torch.cuda.max_memory_allocated` and confirmed with `nvidia-smi`), while peak system RAM during $k$-means clustering was 3.2 GB (monitored with memory_profiler). A fine-grained timing breakdown is as follows: MiniLM content embeddings (1.9 min), BERT style embeddings (5.6 min), two sentiment passes (9.2 min total), an additional MPNet pass for the multi-model variant (12.3 min), and CPU-side clustering/group selection (7.4 min). Thus, Improvement 2 may potentially even fit on a single commodity GPU ($< 4$ GB VRAM) and completes in well under one hour—orders of magnitude for small to medium sized datasets, making it non-problematic to adopt.

## B  Sample Constitutions

In the following we showcase three representative constitution families produced by the second improved version.

---

**Realistic (10 clauses)**

1. Select the response that addresses legal implications.

2. Select the response that avoids unnecessary assumptions.

3. Select the response that discourages harmful actions.

4. Select the response that maintains a neutral tone.

5. Select the response that seeks clarification.

6. Select the response that maintains ethical standards.

7. Select the response that is more concise and direct.

8. Select the response that addresses potential harm concerns.

9. Select the response that encourages further dialogue and understanding.

10. Select the response that avoids encouraging illegal activities.

---

**Semi-Synthetic (10 clauses)**

1. Select the response that provides detailed step-by-step instructions.

2. Select the response that uses consistent and correct language.

3. Select the response that provides specific information.

4. Select the response with higher confidence level.

5. Select the response that includes detailed code examples.

6. Select the response that provides a practical solution.

7. Select the response that maintains a neutral and factual tone.

8. Select the response that uses engaging and vivid language.

9. Select the response that provides clear logical reasoning.

10. Select the response that directly addresses the question.

**Synthetic (5 clauses)**

1. Select the response that provides accurate and corrected information.

2. Select the response that avoids redundancy in advice.

3. Select the response that provides a more concise explanation.

4. Select the response that promotes respect and kindness.

5. Select the response that discourages illegal activities.

