# OpenReview forum: "Decoding Human Preferences in Alignment: An Improved Approach to Inverse Constitutional AI"
_TMLR — Rejected by TMLR_

### Review · Reviewer_xYui · 2025-05-07

**Summary Of Contributions:**

The paper delivers two lightweight yet effective upgrades to ICAI, boosting rule-extraction accuracy across datasets. It pioneers a content/style/sentiment embedding-delta clustering scheme that generalises beyond the case study, and backs its claims with a three-tier experimental suite. Finally, it opens the door to leveraging preference scores for future alignment work.

**Audience:**

Yes

**Broader Impact Concerns:**

This work does not raise any ethical or societal concerns.

**Claims And Evidence:**

Yes

**Requested Changes:**

-  Provide resource-usage statistics: Append GPU-hours, peak VRAM, and wall-time for Improvement 2.
- Conduct human readability study: Sample extracted rules, obtain ratings from multiple annotators on readability and consistency, and report aggregated scores.
- Add quantitative peer comparison: Benchmark against Dromedary and PRELUDE under identical settings, reporting both performance and cost.

**Strengths And Weaknesses:**

### Strengths
- Clear problem statement — the paper pin-points how ICAI over-fits to single examples and tackles it head-on.
- Concise methodological contribution — Both enhancements operate at the prompt-engineering and clustering layers, leaving training pipelines untouched and facilitating industrial adoption.
- Innovative multi-facet embedding delta — Separating content, style, and sentiment embeddings before K-Means clustering yields finer-grained preference rules.

### Weaknesses
- Substantial computational overhead — Triple embedding passes and repeated clustering may be prohibitive at scale; runtime and memory statistics are absent.
- Limited interpretability assessment — Evaluation relies exclusively on LLM-based metrics without human readability or consensus studies.
- Missing quantitative baselines — Comparisons with contemporaneous methods such as Dromedary and PRELUDE are qualitative only; no unified benchmark analysis is provided.

---

> ### Author Response · Authors · 2025-07-02
> **Response to Reviewer xYui**
>
> We thank the reviewer for their review of our work, especially regarding the strengths of our proposed method. We would like to address the reviewer’s requested changes in the following:
>
> 1.
> We provide a short paragraph on the requested statistics in our most recent revision. We find that the GPU-hours and RAM requirements are actually less demanding than they may appear from our description, making the method feasible for small- to medium-sized datasets, even when using small GPUs (< 4 GB VRAM).
>
> 2.
> Due to resource constraints, we are unable to provide a full human-readability study; however, we believe that the quantitative evaluation with regard to the input preference set is arguably more meaningful. Namely, it is not feasible for human annotators to read or even remember large parts of the full dataset, so it would be difficult for them to give a meaningful estimate of rule quality. While we agree that factors such as readability and consistency between different rules constitute an important factor, they are not a primary target of our proposed method. In addition, it would be easy to circumvent proper scoring in such a setup by simply prompting the employed language models to reformulate poor candidate rules or pay special attention to the readability of generated candidates. In a similar manner, while it is generally desirable for a constitution to be self-consistent, this is not a concern for the inverse-constitutional approach, as we merely aim to mirror and concretize the underlying rules used for preference rankings. It is not a direct concern if those rules are inconsistent; in fact, inconsistency may even be desirable in order to spot ambiguous or opposing preference rankings in a given dataset and subsequently address such issues. For these reasons, we believe the current evaluation pipeline sufficiently supports our claims.
>
> 3.
> We appreciate the reviewer’s insight regarding works closely related to our proposal. However, we believe it would be ambiguous or misleading to set up benchmarking against them. We have added further clarification on this in our related-works section, but would like to give brief reasoning here as well. Both PRELUDE and Dromedary are alignment techniques. This is a significantly different target from our goal, which is to extract and formalize underlying principles or rules of preference/ranking datasets. While we perform a brief experiment using AlpacaEval in order to check the potential for pure instruction-tuning using these extracted rules, we do not claim any improved performance or utilization as an alignment technique at all. We clearly state, “Based on our limited evaluation, it is unclear whether constitution-based prompting provides a viable alternative to traditional fine-tuning methods” in Section 3.3 and would like to stress again that the improved ICAI algorithm is not an alignment technique. However, we see potential synergistic application of PRELUDE or Dromedary with our technique-for example, by adapting the extracted principles into longer instructions potentially usable for the Self-Instruct method employed by Dromedary-but this goes beyond the scope of our current submission.
>
> We remain open for any further comment and again thank the reviewer for his insightful comments.

---

### Review · Reviewer_2R7F · 2025-05-25

**Summary Of Contributions:**

This paper aims to improve Inverse Constitutional AI (ICAI) algorithm for extracting constitutional principles from human preference datasets. Traditional LLM alignment methods like RLHF and DPO rely on implicit principles, while Constitutional AI (CAI) uses explicit rule-based frameworks. The authors enhance ICAI’s principle generation, clustering, and embedding processes by introducing more generalizable prompts, centroid-based clustering, and multi-dimensional embeddings (content, style, sentiment). Experiments on synthetic, semi-synthetic, and real-world datasets show modest improvements in preference regeneration accuracy and constitution similarity scores compared to the baseline ICAI. However, the study acknowledges limitations in scalability, ambiguous rule extraction, and underwhelming performance in instruction-following evaluations (e.g., AlpacaEval).

**Audience:**

Yes

**Claims And Evidence:**

Yes

**Requested Changes:**

Please see the Weaknesses.

**Strengths And Weaknesses:**

**Strengths:**
1. This paper identifies weaknesses in the original ICAI algorithm, such as overfitting to specific samples and reliance on random clustering.
2. Using distinct embeddings for content, style, and sentiment adds nuance to principle extraction, addressing different facets of alignment.


**Weaknesses:**
1. The novelty is limited, and the contribution is incremental. The proposed improvements (generalizable prompts, centroid clustering, multi-embeddings) are incremental and do not introduce new technical methodologies.
2. The enhanced pipeline (Improvement 2) requires triple embedding and clustering, increasing computational latency and resource costs. The base ICAI’s sequential prompting already limits scalability, and the improvements exacerbate this without viable solutions.
3. The algorithm struggles with datasets featuring subtle or context-dependent preferences (e.g., varying preferences for reply length in different tasks), leading to ambiguous or over-specific principles.
4. Experimental results are not convincing. Please conduct experiments on more LLMs (e.g., Qwen and Mistral) and more datasets (e.g., instruction following and 3H-related datasets.).

---

> ### Author Response · Authors · 2025-07-02
> **Response to Reviewer 2R7F**
>
> We thank the reviewer for their comments on our work and would like to address the listed weaknesses and requested changes as follows.
>
> 1.
> The reviewer is correct in noting that our work is incremental, but we do not believe this is a significant weakness. We set out to address several clear shortcomings of the original ICAI algorithm and to expand it in ways that, in our view, improve generalizability and interpretability. We further believe that distinguishing different preference origins before clustering is a novel step; we have not found prior work that does so, and we are therefore surprised to see this criticised as lacking novelty. We do, however, understand the comment with respect to other changes, such as stronger prompts, non-random sampling, and improved aggregation order. In sum, we claim incremental—but meaningful—improvements over ICAI, and our methodology supports that claim.
>
> 2.
> It is true that triple embedding increases cost and thus harms scalability. As the reviewer notes, poor scalability is already an inherent limitation of the ICAI approach; tackling that was not the goal of this work. We have added a brief explanation of the resource impact of Improvement 2 in the latest revision. While the impact is certainly not negligible, it should not be overstated. Exploring whether ICAI can operate on smaller dataset samples is a promising avenue for future work, but we feel it lies beyond our current scope. We do not claim improved scalability and explicitly acknowledge the downside of triple embedding; we would be happy to add a short discussion if the reviewer believes it would strengthen the paper.
>
> 3.
> We agree that ICAI struggles with datasets featuring very subtle or ambiguous differences. After reviewing several such preference pairs ourselves, we found that even humans would have difficulty deriving clear underlying preferences, so it is unsurprising that ICAI also struggles. Nonetheless, we report a slight improvement over the original ICAI in this setting, and we hope that our additional evaluation on realistic preference datasets provides value not addressed in Findeis et al.
>
> 4.
> The dataset discussed in point 3 is the original Helpfulness and Harmlessness dataset, which we believe we cover sufficiently. While instruction following is an important alignment target with rich dataset support, it is not well suited to methods that extract constitutions: most instruction tuning is done via supervised fine-tuning, without preference pairs, which ICAI requires. Moreover, the objective of instruction following could arguably be captured by a single rule, making clustering unnecessary.
>
> We thank the reviewer again for their feedback. As noted, this is an incremental work that provides modest improvements and adds evaluations on previously untested dataset domains. Despite limited novelty, we believe our findings—both in evaluation and in rule-extraction methodology—offer useful insights to the research community.

---

> > ### Comment · Reviewer_2R7F · 2025-07-21
> >
> > Thank you for the response. However, Concerns 1-4 have not been addressed. I recommend rejecting this paper.

---

> ### Author Response · Authors · 2025-07-21
> **Response to Reviewer 2R7F**
>
> We respect the reviewers decision, but we would like to clarify once more why we believe to have addressed the listed concerns. First we would like to note that the reviewer does not distinguish between weaknesses and requested changes. It is difficult for us to understand the desired changes or additions to the manuscript without this. Namely, the reviewer criticizes the added latency and resource costs for improvement 2. As we noted in our initial response, we added further content acknowledging this and also performed further calculations regarding the added runtime for this step. Since it only concerns the clustering in a step that precedes the language model prompting, there is no actual added cost incurred by the used language models. In a similar way, we aimed to clarify that the clustering step is novel and therefore constitutes a new technical methodology, which was criticized in concern 1. As we further mentioned, the difficulty with real-life datasets featuring very subtle, or more varied preferences is inherent to all inverse constitutional methods, see for example the paper by Findeis et, al in Appendix F3 or respectively Figure 7 which only slightly outperforms the baseline in agreement. However, it is not true that resulting constitutions are " ambiguous or over-specific", but we recognize that this may have been unclear in our initial version, or our previous response. In order to showcase the quality of constitutions generated in realistic settings, we have added sample constitutions for all three settings in the appendix in our latest revisions. The perceived lack of performance when reproducing user preference does not stem from a flawed constitution, but rather from a high base difficulty of verbalizing/compressing such realistic latent preferences. In fact, our introduced improvements address the very issue of ambiguous or over-specific principles in improvement 1 as we discuss in Section 2.
> Regarding the last point, we have already mentioned our concern in applying ICAI to the requested datasets, and argue that the presented settings suffice to give a fairly complete overview of relevant alignment settings.
> It is difficult for us to adopt the reviewers stance that none of the concerns have been addressed whatsoever without further explanation and we hope that this additional comment serves to communicate this difficulty.

---

### Review · Reviewer_Jcqs · 2025-06-21

**Summary Of Contributions:**

The paper proposed to improve Inverse Constitutional AI (ICAI) algorithm for better extracting interpretable constitutions from preference datasets, from perspectives from the principle generation, clustering, and embedding processing. With the above techniques, the new ICAI approach can facilitate the performance of extracting principles on both human-labeled and synthetic data.

While technically sound, this work suffers from limited novelty, marginal practical improvements, and inadequate validation of its core claims.

**Audience:**

No

**Broader Impact Concerns:**

There is no further social impact or ethical concern or of this paper.

**Claims And Evidence:**

No

**Requested Changes:**

1. This paper needs to clarify the differences or novelties compared to [1], maybe adding a table to contrast the methods, such as embedding types, clustering methods, and principles sampling.
2. Given the AlpacaEval results, the authors may claim this work the work as "enhancing interpretability of preference datasets" instead of a "competitive alternative to fine-tuning".
3. Current evaluation methods/tools are LLM-driven, CAI should involve more human annotating to strengthen the faithfulness. Besides, faithfulness-related datasets can be introduced to evaluate.
4. The proposed methods should get rid of LLMs (e.g., GPT-4o) as much as possible.
5. The authors should validate ablation studies of the impact of distinct types of embeddings (content, style, and sentiment), and the selection of pre-trained/fine-tuned embeddings.
6. For reproduction of this work, this paper should provide more implementation details, such as data sampling, clustering settings, hyper-parameters, etc.

**Strengths And Weaknesses:**

**Strengths**
1. This paper pointed out some critical limitations in ICAI, such as overly specific principles, random cluster sampling.
2. Empirical results in the given Tables demonstrated the effectiveness of the algorithm optimization.
3. The finding of explicit ratings boost accuracy (+3%) is interesting and valuable, positing a promising but underexplored direction.

**Weaknesses**
1. The so called "Improved CAI" primarily combined known techniques: prompt engineering for generalization and centroid-based cluster sampling, and the multi-embedding approach (content/style/sentiment) parallels prior works, for example [1].

[1] Inverse Consitiutional AI. Timothy H. Kostolansky. 2024.

2. No evidence for "competitive alternative to fine-tuning" (the last sentence in Section 3.3), as the AlpacaEval results in Table 3 show no improvement in win rates and standard error.
3. This work employed GPT-4o to generate "chosen" outputs (in Figure 2) risks bias, however, the extracted principles may reflect GPT-4o’s inside biases, not true preferences, making the later experimental results unfaithful. Additionally, similarity scoring with Llama3.1-8B-Instruct in Table 2 shows poor discriminative power (GT vs. GT: 5.8/10), and the over-reliance on unreliable metrics pervades the evaluation.
4. Some practical constraints were not discussed in this paper, such as the scalability issues were not with solution proposals, and processing 3 types of embeddings is prohibitive for larger scale datasets.

---

> ### Author Response · Authors · 2025-07-02
> **Response to Reviewer Jcqs**
>
> We thank the reviewer for their thorough critique and address each point below.
>
> Weaknesses
>
> 1.
> The reviewer correctly notes prior work on Inverse Constitutional AI (ICAI). We cite the original paper by Findeis et al. extensively and list Timothy H. Kostolansky’s Inverse Constitutional AI (2024) in the related-works section. Kostolansky clusters before principle generation and does not aim to produce a minimal representative constitution, whereas our work builds on the Findeis pipeline. Neither prior work introduces multi-embedding clustering; that is a novel contribution of our paper. We will gladly add further citations wherever the reviewer feels they are missing.
>
> 2.
> We acknowledge that our results do not show constitution-based in-context alignment to be competitive with fine-tuning. Indeed, Section 3.3 concludes: “Based on our limited evaluation, it is unclear whether constitution-based prompting provides a viable alternative to traditional fine-tuning methods.” The AlpacaEval experiment is exploratory; it highlights potential future directions, such as combining extracted principles with Dromedary or RL-CAI. Because competitiveness was not our central claim, we regard the unchanged win rate not as a weakness but as clarification of the current limits of instruction-only tuning.
>
> 3.
> The reviewer raises an important concern about GPT-4o bias in the synthetic setting. Owing to the dataset’s small size, we manually audited every generated chosen string and confirmed that each rewrite follows the ground-truth principle without obvious model bias. Including a baseline without extracted rules helps to control any residual bias. Regarding Table 2, the similarity score of 5.8/10 for ground truth versus ground truth is lower than ideal; however, the same scoring rubric is applied to all methods, so the relative comparisons remain valid.
>
> 4.
> We discuss the added cost of triple embeddings and provide resource metrics in the latest revision. Scalability is a known limitation of ICAI, and addressing it was outside our scope, but our new numbers quantify the overhead. Investigating subsampling or more efficient embeddings is an interesting direction for future work.
>
> Requested changes
>
> 1.
> Our Methodology section first recaps the existing ICAI pipeline, then lists its weaknesses and our modifications. Because each change is laid out step by step in detail, we believe an extra comparison table is unnecessary; however, we will add one if the reviewer feels specific points remain ambiguous.
>
> 2.
> We do not claim competitiveness with fine-tuning; the AlpacaEval study illustrates present limitations. If any phrasing still suggests otherwise, please indicate where and we will revise it.
>
> 3.
> We agree that human evaluation is ideal. Unfortunately, meaningful human annotation would require raters to internalize many preference pairs, which is impractical given our resources. We reviewed potential faithfulness datasets such as TruthfulQA-Faithful, FaithDial, and Q2, but they do not contain pairwise rankings, which our method requires. Our evaluation therefore focuses on dataset-generation settings (synthetic, semi-synthetic, realistic) rather than specific task targets.
>
> 4.
> Where feasible, we substitute embedding models for full LLM calls—for example, in our multi-embedding step. Beyond that, LLMs are intrinsic to ICAI. Eliminating them entirely would amount to redesigning the base algorithm rather than improving it, which is beyond the scope of this work.
>
> 5.
> We will provide an ablation study on the individual embedding types. We note, however, that the motivation for the content-/style-/sentiment split is qualitative as well as quantitative.
>
> 6.
> We plan to release the full implementation and generated datasets upon acceptance; these will contain all relevant settings for the reported experiments.
>
> We hope to have addressed the reviewers concerns with our response and are happy to indulge in further discussion. We will further let the reviewer know when we have finished the ablation study from 5. We invite the reviewer to revisit his answers to the questions of claims and evidence, as well as audience. We are pleased to share results in our paper that suggest an incremental improvement over the traditional ICAI algorithm, The reviewer also points out some further strengths themselves. In particular, we have designed our evaluation in a fashion we see as more suitable, as compared to the original ICAI works. Thus, we hope to have at least one relevant claim with our work, and provide evidence to support it. In addition, we believe that there is an audience which would be highly interested in work in this direction. Value-based alignment seeks a lot of attention in recent years, and despite that, the reverse compression of given preference data is fairly unexplored. We are happy to learn more about the reviewers perspective and how we may adhere to it.

---

> ### Author Response · Authors · 2025-07-11
> **Response to Reviewer Jcqs**
>
> We have included the requested ablation study in the latest revision of our work in subsection 3.3. We thank the reviewer again for suggesting this addition, as we agree that it strengthens the claim of improvement 2. We find that inclusion of all three embedding models strengthens performance as compared to singular embedding for all three datasets, and for two datasets when comparing against a selection of two-embedding model combinations. We hope that this addition may further underline the support of our claims by the presented experiments, and potentially motivates the reviewer to reconsider his initial judgement that our work does not provide meaningful claims and evidence.

---

### Review · Reviewer_PnKu · 2025-06-30

**Summary Of Contributions:**

The above text revolves around the alignment methods of Large Language Models (LLMs). The core is the research on the Inverse Constitutional AI (ICAI) algorithm. The contributions are mainly reflected in the following aspects:

1、Aiming at the deficiencies of the Inverse Constitutional AI (ICAI) algorithm, it is committed to improving its performance in extracting constitutional principles from pairwise preference human datasets, solving problems in generality and sampling, etc. It also attempts to use various embeddings to group preference pairs, so as to more accurately generate constitutional principles that reflect dataset preferences.

2、It clearly focuses on three key research questions, including how to improve the existing constitutional extraction method based on pairwise preference human datasets, what the key use cases and implications of representative constitutions are, and whether the generated constitutions can become a competitive alternative to traditional fine-tuning methods, clarifying the direction for research in this field.

3、Evaluate the improvements in different scenarios such as synthetic, semi-synthetic, and real data, and report the improvements compared to the benchmark ICAI algorithm, verify the improvement effects, help promote the development of large language model alignment methods, and explore new paths beyond traditional fine-tuning.

**Audience:**

Yes

**Broader Impact Concerns:**

There is no further social impact in this paper.

**Claims And Evidence:**

Yes

**Requested Changes:**

See weakness.

**Strengths And Weaknesses:**

Strengths：

1、The article analyzes the deficiencies of Constitutional AI (ICAI), and improves its performance in extracting constitutional principles from pairwise preference human datasets, so as to generate constitutional principles that more accurately reflect the preferences of the datasets.

2、Evaluate the improvements in different scenarios such as synthetic, semi-synthetic, and real data, and report the improvements compared to the benchmark ICAI algorithm, verify the improvement effects, help promote the development of large language model alignment methods, and explore new paths beyond traditional fine-tuning.

Weakness：

1、The article argues that the previous methods based on the reward - punishment approach cannot explicitly supervise the model training, while the approach of establishing core values can better help understand the changes induced by the alignment process. Please conduct some qualitative and quantitative analyses.

2、The number of references cited in the article is a bit small. Please conduct additional research and discuss the similarities and differences with the method in this paper.

---

> ### Author Response · Authors · 2025-07-01
> **Response to Reviewer PnKu**
>
> We thank the reviewer for his insightful comment and seek to address the mentioned concerns with our first revision.
> 1.
> We have reflected carefully on the request for “qualitative and quantitative analyses.” Below we clarify why we believe the current manuscript already supplies both, and thus no additional studies are required.
>
> Clarification of our core claim
> Our paper argues that reinforcement-learning–based alignment methods (e.g., RLHF, DPO) learn a reward function in latent weight space, so the supervision signal is not human-readable. In contrast, our constitution-extraction pipeline supervises the model with natural-language rules, making each training-induced change explicitly interpretable. The distinction is laid out in the Introduction, where we also note that latent-reward methods risk silently encoding majority-group biases, whereas an explicit constitution makes such biases visible and editable. The existence of such biases is a first qualitative example for the lack of explicit supervision in RL-Alignment, since they were not recognized prior to actually applying the training.
>
> Quantitative evidence already in the paper
> Section 3, Tables 1 constitute the requested quantitative analysis:
>
> Preference-regeneration accuracy (Table 1, § 3.1) – Our extracted constitutions alone predict unseen human preferences with up to 94 % accuracy, significantly above both the ICAI baseline and an orthogonal-rule control, demonstrating that the explicit rules capture the latent reward signal numerically. In turn, we argue that, using an aggregated representation of such preferences, following alignment techniques, such as Dromedary, which merely rely on instruction-tuning, facilitate explicit supervision.
>
> Qualitative interpretability:
> Section 2.2 (Methodology) and the Discussion give concrete rule snippets extracted by our method—for example, the principle “Select the response that engages with the user’s interest in Sahawiq.”—and explain how such rules expose narrow or majority-driven preferences embedded in the dataset. The fact that such rules are pruned in the ICAI algorithm, whereas this cannot be controlled or audited in the Reinforcement Learning case, is an examplatory case of why ICAI enables explicit supervision.
> 2.
> We appreciate the reviewers insight into the integration of our work into the relevant research context. We agree that this is of significant importance and have performed further research on relevant literature. Our initial paper presented a more narrow view on works exploring the reverse perspective of identifying insights from preference data, which we have expanded now to relevant insights into the potential goals of alignment, as well as additional applied techniques like Moral Graph Elicitation or Sparrow.
>
> We hope to have addressed the reviewers concerns and are open to further discussion, should the reviewer have further questions or comments.

---

> > ### Comment · Reviewer_PnKu · 2025-07-07
> >
> > The authors have addressed most of my concern and I recommend to accept the paper.

---

### Author Response · Authors · 2025-07-11
**General Comment**

We would like to sincerely thank all reviewers and the action editor for their thoughtful feedback throughout the review process. As we approach the conclusion of the discussion phase, we have made a number of important revisions to strengthen our submission. These include the addition of detailed runtime and hardware specifications, an expanded literature review, clarified discussions on how our approach draws from and differs from prior work on Inverse Constitutional AI (ICAI), and a new ablation study examining the impact of different embedding model combinations. We believe these changes have substantively improved the clarity, rigor, and completeness of our paper, and we are grateful to the reviewers for highlighting these points.

We would also like to reiterate the intended contributions of our work. While our paper does not claim to redefine ICAI or introduce an entirely new algorithmic paradigm, it offers a meaningful extension to existing approaches. In particular, we identify shortcomings in prior ICAI methods and introduce novel components, most notably, a multi-embedding stage, that consistently improve performance across all evaluated settings. We acknowledge that these gains are incremental, but they are robust and well-supported by empirical results.

Beyond performance, our work contributes new perspectives by evaluating ICAI techniques on underexplored settings, such as semi-synthetic datasets. For example, we explore the utility of incorporating preference scores where available (e.g., in UltraFeedback), and we highlight the limits of using generated constitutions for in-context instruction tuning, underscoring that ICAI may be more impactful as a tool for interpretability and bias analysis than for direct performance gains.

We hope that the reviewers recognize the relevance and merit of our contributions, and appreciate the additional insight and clarity added through our revisions. We remain open for any further discussion.

---

### Decision · Action_Editor_ev14 · 2025-08-12

**Recommendation:** Reject

**Additional Comments:**

While the core concept of ICAI shows promise, the current manuscript requires substantial revisions across multiple dimensions:

**Major Issues to Address**

1. Expand the literature review and provide clearer context
2. Conduct comprehensive comparisons with constitutional approaches like InverseCAI
3. Move beyond the LLM-as-a-judge paradigm to include human evaluation and rule-based evaluation
4. Investigate potential "over-principled" phenomena and discuss applicability limitations
5. Include detailed implementation specifications and resource requirements

**Technical Concerns**

The paper's potential narrow algorithmic applicability and potential over-principled phenomena on certain preference datasets require empirical investigation. The authors may identify dataset types where ICAI might struggle and provide honest assessment of method limitations.

A comprehensive revision addressing these fundamental issues would be necessary for future consideration.

**Audience:**

Yes

**Audience Explanation:**

The core idea behind ICAI - learning interpretable principles from preference data for alignment - addresses an important problem in the field of AI alignment and interpretability. The TMLR audience, which includes researchers working on machine learning safety, alignment, and interpretability, would likely find this research direction valuable. However, the current execution and presentation significantly limit the impact and reliability of the findings.

**Claims And Evidence:**

No

**Claims Explanation:**

The paper introduces ICAI, a method for learning hidden principles from preference pairs to enhance alignment performance and interpretability. However, several critical issues undermine the evidence supporting its claims:

1. The experimental evidence is inadequate, with unconvincing results on AlpacaEval and evaluation limited to the LLM-as-a-judge paradigm. The lack of human evaluation or other established benchmarks weakens the validation.
2. The paper fails to provide adequate experimental comparisons with existing methods, particularly missing direct comparison with related approaches like InverseCAI [1].
3. The submission lacks crucial implementation specifics and computational resource consumption analysis, which are essential for reproducibility and proper evaluation.
4. The paper lacks proper ablation studies, particularly on embedding selections, making it difficult to justify design choices.

In general, there are two reviewers who reject, one reviewer leaning toward accept, and one reviewer who accepts. Specifically, among them, one reviewer raised significant concerns, and claimed that the authors' rebuttals cannot address the concerns raised. I carefully read the author's rebuttal, and I think the authors need to support their claims with more solid empirical work.

Given the significant empirical research limitations in this paper, it would be difficult for the authors to address these concerns adequately in the current round. The authors are strongly encouraged to carefully address the concerns raised by reviewers and provide more robust empirical evidence for the revised version.

[1]: Inverse Constitutional AI: Compressing Preferences into Principles

**Resubmission Of Major Revision:**

The authors may consider submitting a major revision at a later time.